



# Knowledge-inspired fusion strategies for the inference of PM2.5 values with a Neural Network

Matthieu Dabrowski[1], José Mennesson[2], Jérôme Riedi[3], Chaabane Djeraba[1], and Pierre Nabat[4]

[1]Centre de Recherche en Informatique Signal et Automatique de Lille, Univ. Lille, CNRS, Centrale Lille, UMR 9189 CRIStAL, F-59000 Lille, France.
Correspondence : Matthieu Dabrowski (matthieu.dabrowski@univ-lille.fr), Chaabane Djeraba (chaabane.djeraba@univ-lille.fr)
[2]IMT Nord Europe, Institut Mines-Télécom, Univ. Lille, Centre for Digital Systems, F-59000 Lille, France.
Correspondence : José Mennesson (jose.mennesson@imt-nord-europe.fr)
[3]Univ. Lille, CNRS, UMR 8518 Laboratoire d'Optique Atmosphérique (LOA), 59000 Lille, France
Correspondence : Jérôme Riedi (jerome.riedi@univ-lille.fr)
[4]Centre National de Recherches Météorologiques - Météo-France, Groupe de Météorologie Grande Echelle et Climat, , Toulouse, France.
Correspondence : Pierre Nabat (pierre.nabat@meteo.fr)

**Abstract.** Ground-level concentrations of Particulate Matter (more precisely PM2.5) are a strong indicator of air quality, which is now widely recognized to impact human health. Accurately inferring or predicting PM2.5 concentrations is therefore an important step for health hazard monitoring and the implementation of air quality related policies. Various methods have been used to achieve this objective, and Neural Networks are one of the most recent and popular solutions.

In this study, a limited set of quantities that are known to impact the relation between column AOD and surface PM2.5 concentrations are used as input of several networks architectures to investigate how different fusion strategies can impact and help explain predicted PM2.5 concentrations. Different models are trained on two different sets of simulated data, namely global scale atmospheric composition reanalysis provided by the Copernicus Atmospheric Monitoring Service (CAMS) as well as higher resolution data simulated over Europe with the Centre National de Recherches Météorologiques ALADIN model.

Based on an extensive set of experiments, this work proposes several models of knowledge-inspired Neural Networks, achieving interesting results both from the performance and interpretability points of view. Specifically, novel architectures based on BC-GANs (which are able to leverage information from sparse ground observation networks) and on more traditional UNets, employing various information fusion methods, are designed and evaluated against each other. Our results can serve as baseline benchmark for other studies and be used to develop further optimised models for the inference of PM2.5 concentrations

from AOD at either global or regional scale.

## 1 Introduction

Particulate Matter (PM2.5), defined as fine airborne particles with an aerodynamic diameter of less than 2.5 micrometers, serves as a critical indicator of air quality. PM2.5 levels are strongly associated with adverse health outcomes, including respiratory and cardiovascular diseases (Bose et al., 2015; Madrigano Jaime et al., 2013; Neophytou et al., 2014). The Global Burden of



Disease study has recognized air pollution as the fifth leading risk factor for mortality worldwide (Cohen et al., 2017). Accurate estimation and prediction of PM2.5 concentrations are therefore essential for effective health hazard monitoring.

Research on the health effects of PM2.5 is fundamental for the development of air pollution management strategies. Access to air pollution exposure data is also critical for assessing the negative health impacts of ambient PM2.5. Historically, regional and national ground monitoring networks have been the primary sources for PM2.5 data. However, the establishment and
maintenance of such networks are costly, especially on a large scale, and may not be prioritized in some countries. (Martin et al., 2019) report how a substantial portion of the world lacks adequate PM2.5 monitoring, with only 10% of countries having more than three monitors per million inhabitants and 60% of countries not conducting routine PM2.5 monitoring. Furthermore, the scarcity of historical data impedes longitudinal health studies. For instance, China's or India's PM2.5 nationwide monitoring networks were only established respectively in late 2012 and 2015, resulting in a lack of data prior to those dates (Ma et al.,
2019; Dey et al., 2020).

Networks of ground-based sensors monitoring PM2.5 concentration at surface level are instrumental but can only provide information for a few sparse locations. Obtaining complete maps of PM2.5 values from satellite observations is therefore an interesting and important task. Aerosol Optical Depth (AOD), a metric used to indicate aerosol loading in the vertical column, has strong positive relationships with ground-level PM2.5 concentrations (Jill A. Engel-Cox and Haymet, 2004; Wang and
Christopher, 2003; Mukai et al., 2006; Xin et al., 2014). In recent decades, advanced space-borne sensors have provided AOD measurements with broad spatial coverage and high spatial resolution. This has enabled the use of satellite derived AOD products for large scale estimate of mass concentration at ground level through more or less complex AOD-PM2.5 conversion schemes and models (van Donkelaar et al., 2006; Wu et al., 2016; Hu et al., 2014; Chu et al., 2016; Di et al., 2019; Guo et al., 2021; Ma et al., 2022; Gilik et al., 2022).

However, while space-borne observations of aerosol properties, such as AOD and the Ångström exponent, can provide large-scale information, these quantities are not easily nor directly related to PM2.5 concentrations near surface level. This is because the PM2.5-AOD relationship can be a multivariate function of a wide range of influencing factors. At first order, AOD and aerosols properties (fine mode fraction, hygroscopicity) are indeed skillful predictors for near-surface PM2.5 concentration. The literature, however, points to a wide range of parameters that may also contribute positively to PM2.5 statistical prediction.
Meteorological variables (wind speed, height of the planetary boundary layer (HPBL), humidity, temperature, rainfall), surface conditions (albedo, normalized difference vegetation index (NDVI)), distance to the ocean, road infrastructure, population density, elevation or calendar month are regularly considered as useful influencing factors (Lary et al., 2015; Son et al., 2018; Reid et al., 2021; Su et al., 2022).

Among the numerous studies aimed at retrieving PM2.5 concentrations from satellite, we can generally identify three main
categories of methods. The first ones are based on atmospheric chemical transport models (CTMs) and establish a scaling factor between simulated values of AOD and PM2.5 (Lyu et al., 2022; Xiao et al., 2022). This factor can then be transferred to estimate ground level PM2.5 from satellite derived AOD (van Donkelaar et al., 2006; Geng et al., 2015). This method accuracy heavily depends on the scaling factor spatiotemporal variability and has therefore clear limitations if the variability is not properly accounted for and represented by the scaling model. The second set of methods are directly data-driven and aim at



establishing a univariate or multivariate statistical relationship between AOD, other influencing factors and ground-level PM2.5
observed concentrations. While the initial studies proposed to use simple linear or generalized linear regression models, more
complex nonlinear methods, such as neural networks (Gupta and Christopher, 2009) or boosting (Reid et al., 2015), have been
applied later. Machine learning techniques have developed rapidly (Irrgang et al., 2021; Unik et al., 2023) and proved highly ef-
ficient for representing the nonlinear relationships between PM2.5 and multiple variables (Lee et al., 2022). Yet, performances

of machine learning based methods remain eventually affected by the distribution and density of ground stations used to feed
the regression algorithms (Gupta and Christopher, 2009; Li et al., 2017). Finally, a third type of approaches combined physics
based explicit relations between core aerosol properties (size distribution, hygroscopicity, optical extinction efficiency) and
PM2.5 concentrations. While those also rely partly on empirical formulation for establishing some parameters (especially the
link between optical properties and aerosols composition), they tend to provide a better physical interpretability than purely

statistical methods and are also more independent of ground stations observations specifics. Combining the interpretability
advantage of semi-physical empirical models with the strength of machine-learning to improve the accuracy of physical pa-
rameters acquisition, opens a clear path to obtain accurate PM2.5 concentration from satellite observations as illustrated by
(Jin et al., 2023a).

    Machine learning has been increasingly used to develop PM2.5 models and deep learning, in particular deep convolutional

neural networks (DCNN), has recently revolutionized many prediction-related application areas, including diagnostic. Several
recent and extremely thorough review papers provide clear evidence for the exploding number of studies in the field (Ma et al.,
2022; Unik et al., 2023; Zhou et al., 2024) and also illustrate the need for more standardized comparison methodologies and
metrics (Zhou et al., 2024).

    While models tend to perform increasingly well, especially once optimized for a particular region (Chen et al., 2024), they do

not necessarily help understand the relative importance of input parameters on final decision. An old and persistent criticism of
neural networks (NN) among physicists is that they do work often at the expense of hiding physical understanding, especially as
NN based models tend to rely on increasingly complex architectures. Not surprisingly the general growing interest in so-called
"explainable AI" is also echoed in sciences (Beckh et al., 2021), including atmospheric sciences, as the use of deep learning
create paradigm shifts in atmospheric modeling. In that respect, the study by Park et al. (2020) provides a valuable approach to

evaluate model sensitivity to predictors through layerwise relevance propagation (LRP) (Bach et al., 2015) but remains quite an
exception among the ocean of PM2.5 models. Finally while ML actually provides skillful models, there has been little work in
the atmospheric sciences to understand how 2D AOD distribution could actually inform on aerosol properties and be combined
with column properties in order to improve AOD to PM2.5 scaling. While some essential parameters are not easily handled
or predictable (boundary layer height, aerosol type Fine Mode Fraction and aerosol vertical profiles) all depend strongly on

atmospheric dynamics and geographical location which in turn is somehow translated in the 2D AOD distribution. CNN have
shown excellent generalization capability for dealing with input data that has spatial auto-correlation, like images (Szegedy
et al., 2016) and are therefore potentially well suited in order to extract the information on aerosol properties contained in their
spatial distribution (Marais et al., 2020).





Among the three different approaches often used to estimate PM2.5 from AOD observation, we explore here an hybrid method for addressing the scaling approach. We use DCNN (Deep Convolutional Neural Networks) or DC-GAN (Deep Convolutional Generative Adversarial Networks) in order to better capture the spatiotemporal heterogeneity of the PM2.5-AOD relationship. We aim at testing different architectures and information fusion strategies in order to develop a model for PM2.5 which results and performances can be better explained.

In previous work (Dabrowski et al., 2023), the AOD alone is used for the inference of PM2.5, which leads to promising results, surpassing other methods such as Polynomial Interpolation and the Random Forest Machine Learning algorithm. However, other variables (such as the surface-level wind speed and direction, temperature, pressure, humidity and Ångström exponent), could be used as well, as they are known to strongly drive surface PM2.5 concentrations (Unik et al., 2023). We evaluate in the present study if these additional information could enhance the inference performance depending on network architecture and information fusion strategy.

The main contributions of this paper are :

- a study on the interest of several variables (Ångström exponent, wind speed and direction, temperature, pressure, humidity) for the prediction of surface PM2.5 concentration when used jointly with the AOD

- a study on the best type of fusion method to use for the prediction of surface PM2.5 concentration, depending on the variables used as input and on the type of model used

- based on the knowledge from these studies, a model architecture is proposed, along with a selection of additional input variables to use

The insights this study provides and the knowledge it represents help in building an efficient (and knowledge-inspired) model. Indeed, based on a performance analysis, we propose a combination of network architectures that appear most suitable for the application. We note here that our main objective is not to develop an optimized network for a specific application but rather to investigate whether certain types of network architecture or fusion strategies may be more suitable for leveraging information contained in 2D multi-component atmospheric fields for aerosol characterization.

This paper begins with an explanation of the experimental approach chosen to investigate this problem, in section 2. Then section 3 proposes a more in-depth description of the data used. Section 4 is dedicated to describing the models and methods proposed as solutions in this paper. A quick overview of several relevant concepts from the fields of Machine Learning, Deep Learning and Computer Vision is provided in subsection 4.1, followed by a more detailed explanation of the models of interests. Indeed, some other methods are only used as a baseline for comparison. A precise description of our experiments is realised in section 5, along with their results and interpretation in section 6. Finally, section 7 gives an overview of the main findings and proposed solutions deriving from this study.



## 2  Objective and Approach

The purpose of the models designed in this paper is to infer maps representing values of the PM2.5 concentration at ground level, from maps (of the same size) representing values of the AOD in conjonction with maps of other atmospheric variables. To do this, NN such as UNets and GANs are used as their convolutional versions showed an ability to take into account the spatial variability of the data they are being presented with. In the case of Convolutional Neural Networks (CNN) these data mainly take the form of images or matrices.

As further detailed in section 3, we use different aerosol optical properties in addition to AOD and meteorological quantities that are known to caracterize or drive the aerosol concentration as well. These are namely, the wind speed and direction, atmospheric pressure, temperature, relative humidity (all five of these meteorological variables being measured at surface level) and the Ångström exponent.

An important number of experiments are conducted in order to study the impact of these additional variables on the inference
performance for each network architecture. Furthermore, as stated in section 4.3, there exists different strategies to leverage several inputs at the same time and within the same model. In this work, we test three different fusion techniques, namely: feature fusion (FF), decision fusion (DF) and channel concatenation (CC) (also called data fusion). These strategies and their implementation are described in section 4.3. Experiments are performed to identify the best fusion strategy depending on network architecture and available input variables considered for inference of PM2.5.

More classical solutions, such as the kriging method or even polynomial interpolation, are implemented as well to serve as baseline for comparison of inference performances. The expected outcome of this important number of experiment is a performant NN architecture for the prediction of PM2.5 concentration from complete and incomplete maps of the AOD, along with insights on the design process of this type of model.

## 3  Data

In this work, we exclusively use data from simulations (namely from the CAMS and ALADIN models), as it allows us to easily obtain all necessary information. It also maintains the possibility to select or sample data to represent realistic observation scenarii. The CAMS model provides maps representing values of various meteorological quantities and optical measurements, covering the entire world.

The ALADIN model provides the same type of maps, but they cover Europe and the North of Africa instead of the entire
world, and at a higher spatial and temporal resolution than CAMS.

Even though we use simulated data, our objective remains to simulate what could be obtained in a real situation. This is why we come up with a scenario in which a part of the data is simulated, and the most recent part is real, measured data. More precisely, in this hypothetic scenario, optical sensors operating from geostationary satellites (Ceamanos et al., 2021) allow us to obtain AOD values in near real-time. These satellites are, namely, two Meteosat Second Generation (MSG) satellites,
the Himawari satellite, and two Geostationary Operational Environmental Satellites - New Generation (GOESNG). They respectively cover Europe and Western Asia, Eastern Asia, and the Americas. The cumulated coverage of these geostationary





satellites allows to generate complete (as opposed to sparse) maps of the AOD. The PM2.5 concentration values at surface level would be obtained through photometers, Lidar instruments, optical counting sensors or even filters. Each of these sensors can only provide concentration values for its own geographical location. This is why this network of sensors can only provide sparse maps of the PM2.5 concentration.

This means that, for a real use-case scenario, no complete ground truths are available in the measured data. Instead, sparse ground truths are available. In order to reproduce that scientific obstacle, we produce sparse maps of the PM2.5 concentration using the complete ones, by randomly selecting pixels. For a part of the training set, we consider having only access to these sparse maps instead of the complete ones. This was suggested by the authors of (Dabrowski et al., 2023), as it allowed for some level of control over the sparsity of these sparse maps, and therefore allowed for a study of the impact of the sparsity of these maps over the results. We use this method too in order to be able to compare the results of this paper to ours.

The Aerosol Optical Depth, expressed for a wavelength of $550nm$, is our main input and is used systematically. Apart from it, six other quantities that are either routinely observed or modelled can be used as additional inputs. Five of them are meteorological quantities : wind speed and direction, relative humidity, temperature, and pressure, all of which are measured at surface level. These quantities are known to drive aerosols concentration and their size distribution. The last quantity, the Ångström Exponent (AE), is actually an optical quantity that is derived from AOD at two different wavelengths. It characterizes the spectral variation of the AOD and is related to aerosol particle size distribution such that aerosols with a dominance of fine particles will tend to exhibit larger AE. This is therefore an important parameter in aerosol modeling and potentially an important predictor of PM2.5 (Jin et al., 2023b).

Links to the data and code used in this article are available in the code and data availability section, just after section 7.

## 4 Models

### 4.1 Background

The task of inferring maps of PM2.5 at ground level from maps of AOD combined with one or several other variables, can be seen as a regression problem, or as what is known in the field of Computer Vision an "image-to-image translation" problem. Indeed, we want to infer an output image from a different input image (or from a number of them). In the litterature, one can find several methods used to solve this kind of task such as the polynomial interpolation, the kriging method (Matheron, 1963), machine learning (Ho, 2018) and deep learning algorithms (Goodfellow et al., 2020). The most relevant algorithms in our context will be briefly described below. Details can be found in section A.

**Kriging** (Matheron, 1963) is a spatial interpolation and extrapolation method governed by prior covariances. It performs better with important volumes of data, and when estimated values follow a normal distribution. For each inference, a new kriging model is built, which leads to longer inference times compared to other methods used in this paper. This method is described in greater detail in section A.

**UNets** (Ronneberger et al., 2015) represent a type of Neural Networks (NN) architecture, known for its performances, particularly in the field of Computer Vision. The architecture is typically composed of an Encoder and a Decoder, and the main




idea behind UNets is to add skip connections linking the outputs of each layer in the encoder to a corresponding layer in the decoder. This makes it possible to reduce dimensionality without the risk of losing relevant information in the process. Figure 1 gives an example of a model with this type of architecture.

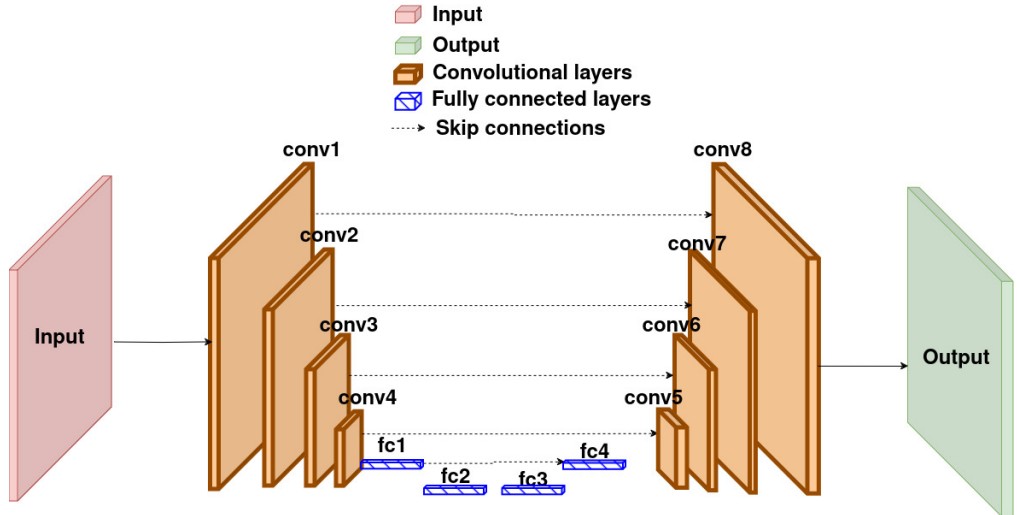

**Figure 1.** Architecture of a UNet with both Convolutional and Feed-Forward layers. Here, the Encoder and Decoder are symmetrical.

**GANs** (Goodfellow et al., 2020) are a type of NN that actually consist of two networks. One is called the generator, and the other is the discriminator. The role of the discriminator is to distinguish between real data and data generated by the generator.

The purpose of the generator is to produce output close enough to the real data that the discriminator labels them as real. These two networks learn competitively: the higher the loss of the generator, the lower the loss of the discriminator, and inversely. This allows such models to show interesting performances in the context of semi-supervised learning. They are also highly efficient for image-to-image translation tasks. The learning process of the type of GAN that will be used in this paper is described in (Dabrowski et al., 2023), in figure 1 of that article.

### 4.2 Relevant Deep Learning models and architectures

Different types of model are implemented, trained and tested on our data. We use as baseline for comparison the Random Forest algorithm (a Machine Learning algorithm), a polynomial interpolation method (of degree 3), and a kriging algorithm as described in 4.1.

Two Deep convolutional neural networks architectures, UNets and BC-GANs (Dabrowski et al., 2023) are the basic compo-

nents of the models we propose in this paper. The first one is a purely supervised UNet, and the second one a semi-supervised BC-GAN as described in 2. The architecture of the generator of these BC-GANs will each time correspond to the architecture of the corresponding purely supervised UNet. As for the architecture of the discriminators, they are described in section B.





These models allow to leverage sparse ground truth when complete ones are unavailable, which increases performance (compared to classical GANs) in the context of semi-supervised learning.

Indeed, we do not have access to complete ground truths for the whole training set. For a part of it, we only have access to sparse ground truths. The authors of (Dabrowski et al., 2023) were mainly interested about how these sparse ground truths and the information they represent could be leveraged in order to ease the training and obtain better results. They proposed a method to leverage those sparse ground truths based on the literature around Physics-Informed Networks, that implied seeing those sparse ground truths as Boundary Conditions (BC), hence the name of BC-informed GAN. The authors of (Dabrowski

et al., 2023) illustrate this method in figure 2 of their article. It includes the design of an additional loss function in order to train the model to respect the BCs. This essentially allows for localised supervision.

### 4.3    Information fusion strategies

All our models have in common that they use as input one or several variables to produce the same type of output. It is therefore necessary to merge these variables together during this process. This also ensures the production of the output makes use of all

these pieces of information.

    Fusion strategies are therefore an important aspect of the architecture definition, and eventually the performance, of the model. They represent different methods that can be applied to leverage several sources of data (several inputs) within the same NN. They can be applied on models such as UNets as well as GANs. There are three main different fusion strategies according to (Mangai et al., 2010), namely data fusion, feature fusion and decision fusion. Sections 4.3.1, 4.3.2 and 4.3.3

describe (respectively) each of these approaches and the way we use them.

#### 4.3.1    Data fusion (channel concatenation)

The general idea behind this strategy is to use several pieces of raw data to build a new, more complete and useful, piece of raw data.

    As our data consists of images, the simplest way to do data fusion is to realise channel concatenation: in other words, to use

our different inputs as if they were different channels of one single image. For this reason, this method is also called Channel Concatenation throughout this paper. This approach if the most straightforward of the three.

    In terms of architecture of our neural networks, this simply implies using more convolution filters. The UNet's architecture with data fusion is illustrated by Figure 2. The GAN's discriminator's architecture with data fusion is illustrated by Figure B1 in section B.

In terms of interpretation, this architecture relies on the local (rather than global) relationships between the different quantities used as input. We believe this architecture to work better if local patterns in one input image correspond to local patterns in other input images.





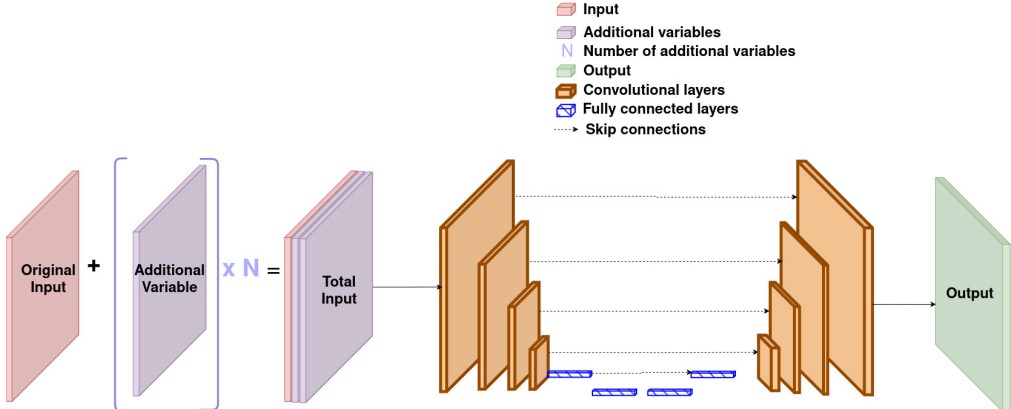

**Figure 2.** Architecture of the UNet with data fusion approach. Also corresponds to the architecture of the GAN's generator.

### 4.3.2 Feature fusion

One of the most common architectures in neural networks for computer vision tasks is the encoder-decoder. The general idea is that the encoder generates what is called a feature. This feature is usually a vector, but can technically also be a matrix (although the idea is for it to be of smaller dimensions than the input). It is supposed to contain all the relevant information from the input with regard to the task at hand. In other words, it describes the input well enough so that only this feature is needed for the task. During the next step, the decoder uses the feature as input and produces the output.

Feature fusion methods are often applied to computer vision tasks, for example when dealing with complex hyperspectral images (Song et al., 2018). The idea behind them is to obtain a feature for each input, and use them to obtain one super-feature (for example by simply concatenating the various features). According to the authors of (Sun et al., 2005), two interesting ways to fuse feature vectors are the serial feature fusion (based on an union vector) and the parallel feature fusion (based on a complex vector), although the same authors actually propose a new method based on canonical correlation analysis (CCA).

This unique feature is then used by the decoder to produce the output. In terms of architecture, this implies having as many encoders as inputs, but only one decoder (since there's only one output). The UNet's architecture with feature fusion is illustrated by Figure 3.

The UNet architecture is a specific type of encoder-decoder, in which a specific type of connections, called skip connections, can be found. It is also often symmetric (in the sense the decoder's layers mirror the encoder's one). After each layer in the encoder, the obtained feature is sent to the corresponding layer in the decoder. This allows for the decoder to have access to several features rather than simply the smallest one.

Implementing feature fusion with a UNet is therefore non-trivial: among all the features obtained for each input, which ones should be sent to the decoder through skip connections ? We choose to apply what we call multiple feature fusion. The principles of feature fusion are applied to feature of each and every scale, and those merged features are sent to the corresponding layer of the decoder through skip connections.




In terms of interpretation, this architecture relies on the global relationships between the inputs. As obtaining features relies on dimension reduction, those features represent the input in a more global way, and do not necessarily represent local patterns. The smaller the scale of these features (and the deeper their corresponding layers are), the truer it is. This architecture relies on the idea that each of the input images contains a global, non-localized piece of information that can be useful in order to estimate the aerosol concentration. Again, the global or non-localized aspect of each of the features actually depends on its

scale or depth in the network.

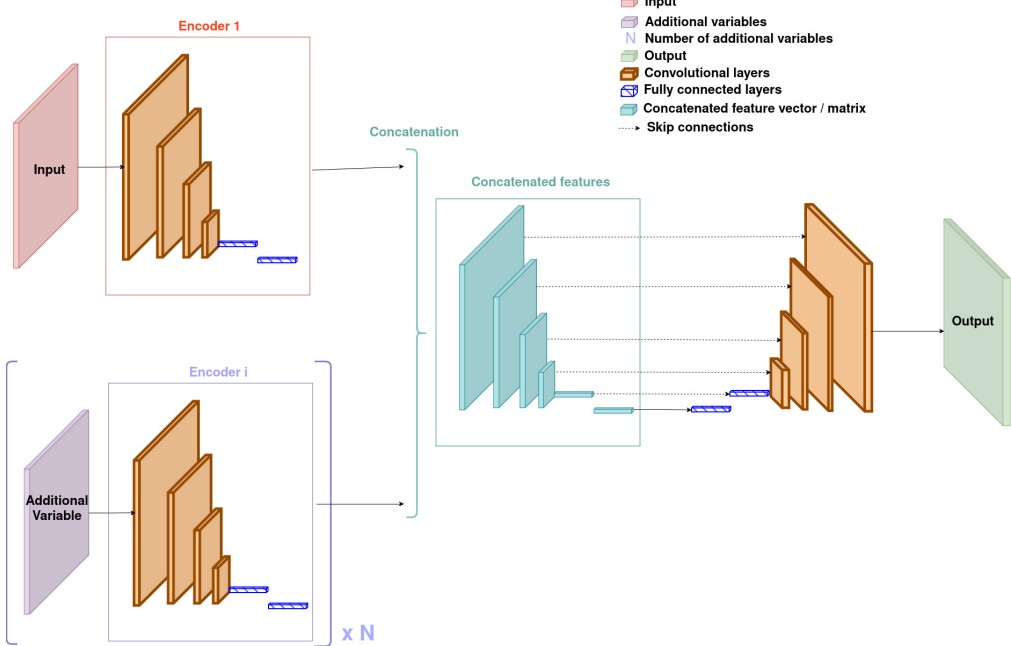

**Figure 3.** Architecture of the UNet with feature fusion approach. Also corresponds to the architecture of the GAN's generator.

The GAN's discriminator's architecture with feature fusion is illustrated by Figure B2 in section B.

### 4.3.3   Decision fusion

The idea behind decision fusion is to use a separate model for each input, obtain an output for each of them, and then merge those outputs together to obtain a final decision, supposedly better. In classification tasks, decisions represent the predicted

class. In regression tasks, like the one considered in this paper, they represent the estimated quantity. Losses are computed using the final output. The backpropagation process takes place through the entire model (and the smaller models that compose it).

There exists several ways to fuse decisions, such as the linear or log opinion pool (corresponding respectively to a weighted sum or product) (Sinha et al., 2008). Voting algorithms can even be used for classification tasks (Sinha et al., 2008). In this



article, a linear opinion pool approach is chosen: we apply a weighted mean of all the outputs to obtain the final one. The weights are learnable parameter, which allows the model to learn which outputs are most relevant.

This principle relies on the idea that each of the inputs can individually be used to produce an estimation of the aerosol concentration, but that these estimation may be flawed, and the best estimation can be obtained through a combination (here a weighted mean) of these flawed estimation. In other words, it is possible to correct the estimation produced with an input
using the estimation produced with an other input. Since the model can learn which inputs are the most relevant to produce the desired output, we expect this approach to provide the best inference when all inputs are used.

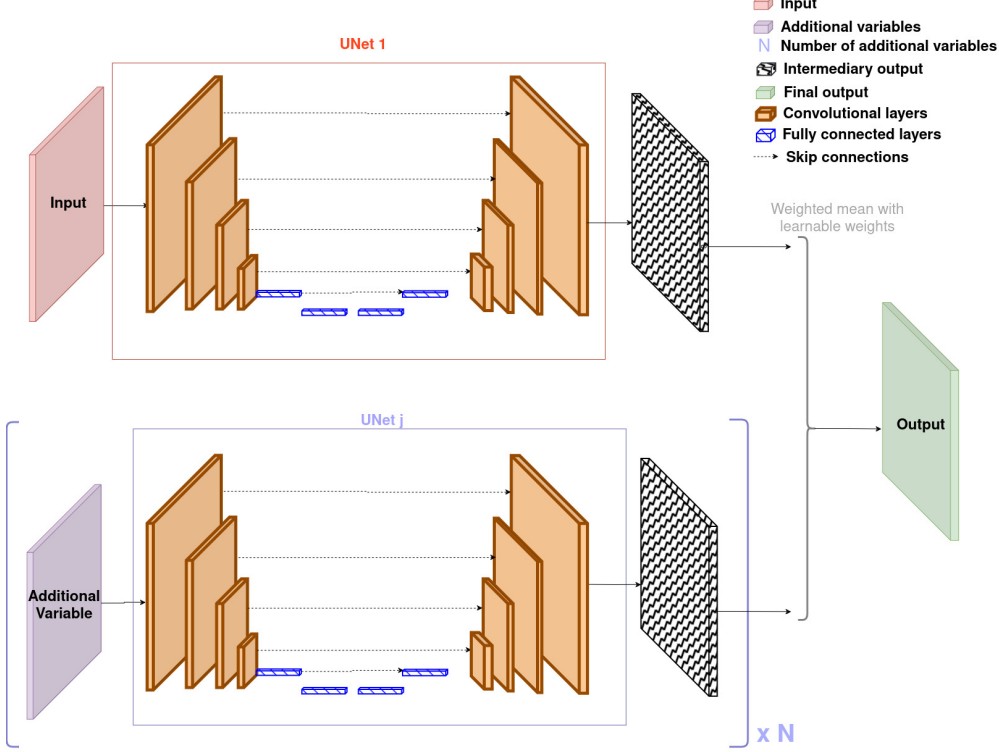

**Figure 4.** Architecture of the UNet with decision fusion approach. Also corresponds to the architecture of the GAN's generator.

The architecture of the GAN's discriminator with decision fusion is, in principle, very similar to the architecture of our UNet with decision fusion. The main difference is the type of output, as the discriminator outputs a single scalar for each iteration while the UNet outputs images. It is illustrated by Figure B3 in section B.

**4.3.4  Hybrid fusion models**

The physical nature of PM2.5 predictors obviously has an impact on the non-linear function linking AOD and PM2.5. While AOD is directly linked to total column aerosol concentration at a given location, surface pressure can indirectly be linked to PM2.5 through accumulation in the atmospheric boundary layer under stable conditions while wind speed can influence





PM2.5 concentration over longer range in space and time. Therefore we can distinguish "state" variables that can directly
link PM2.5 to AOD through an integral expression over the atmospheric column and "indirect predictors" that act on PM2.5
concentrations over different space and time scales. In our current analysis, the Wind variables (speed and direction) stand
out as they describe the atmosphere dynamics while the AOD and Ångström exponent are clearly states variables regarding
the inference of PM2.5 concentration. Humidity, Pressure and Temperature variables, can be considered primarily as states
variables as they strongly impact the particle size distribution through aerosol hygroscopicity but can also indirectly influence
near surface PM2.5 concentrations by favoring accumulation under stable atmospheric conditions or on the contrary by removal
of atmospheric particles through dry deposition or wet scavenging.

The performance of networks and their robustness to noise is known to be impacted by their architecture and network
performances can be improved upon SOTA when the network is well-aligned with the target function (Li et al., 2021). We
hypothesize here that for atmospheric applications, the optimal alignment of network architecture with the target function may
depend on the nature of variables used as input and on the fusion strategy used for merging information carried by those
variables. Through this hypothesis we ask whether there is an advantage in applying different fusion strategies for different
types of input variables.

Based on this insight, we propose two hybrid models, using different fusion strategies depending on the variable considered.
The idea is to use Data Fusion (Channel Concatenation) for the AOD and all other input variables, except for the Wind and
Temperature for which Feature Fusion is used. Figures 5 and 6 describe these two models more precisely. Section 6.5 shows

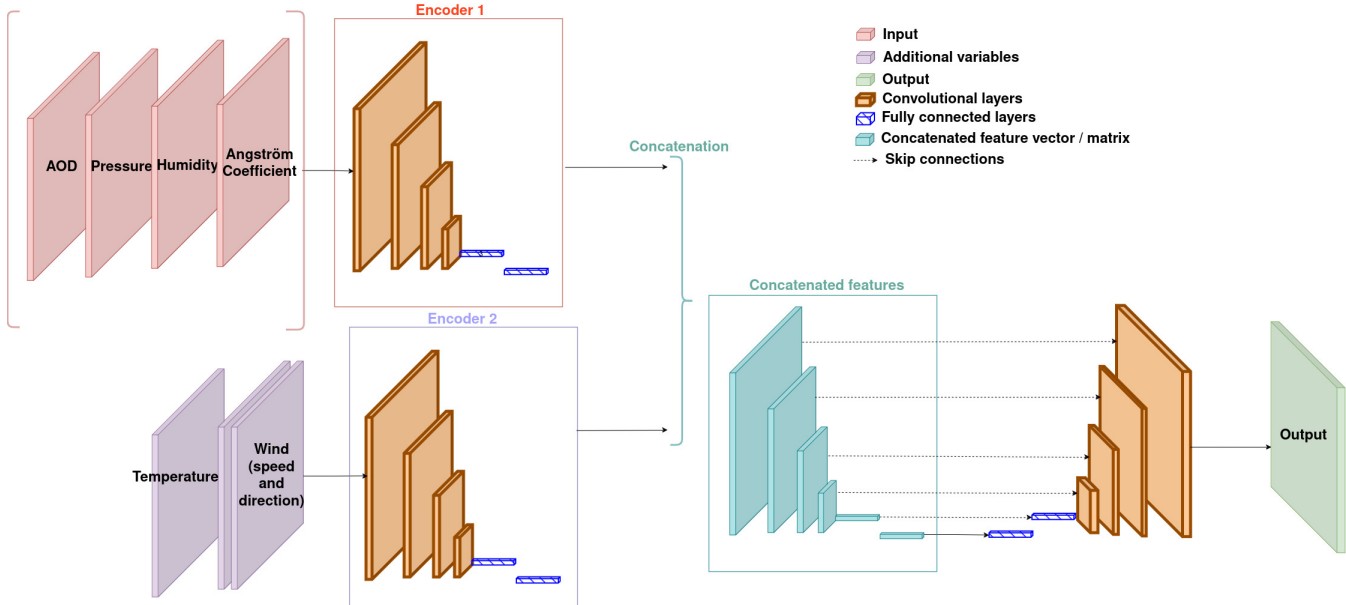

**Figure 5.** The first proposed hybrid model, using both data fusion and feature fusion.

and discusses the results obtained by these models.





**Figure 6.** The second proposed hybrid model, using both data fusion and feature fusion.

## 5  Methodology

In order to find the best way to handle our variety of input quantities, we propose to study the three main fusion approaches described in section 4.3. Each fusion method is experimented on two types of models: a purely-supervised UNet, and a BC-GAN using sparse measurements of the aerosol concentration at ground level as boundary conditions. Experiments on the hybrid approaches proposed in section 4.3 are realised as well.

The goal is also to understand which input quantities have the most important impact on our results, in other words which additional variables actually help our models to better predict PM2.5 at surface level. This is why, for each distinct model architecture, experiments are realised with different combinations of additional variable, to study their impact on the results.

305



## 5.1 Learning and validation protocol

Experiments are conducted on CAMS and ALADIN datasets. AOD is systematically used as input in all our experiments. In addition, six other variables are also considered (wind speed and direction, relative humidity, atmospheric pressure, temperature, Ångström exponent) in order to evaluate their impact on the results. It is important to specify that the wind speed and direction are always used jointly to describe the wind state variable. With this rule in mind, experiments are performed with all possible combinations of these six variables (including using none of them and all of them). The AOD is also used in all cases.

For both CAMS and ALADIN dataset, we always consider the same type of scenario, shown by Figure 7. In this scenario, we have access to a dataset with complete ground truths, corresponding to a period of eleven months. We also have access to a second dataset with sparse ground truths, which can therefore only be used in the context of semi-supervised (as opposed to purely supervised) learning, corresponding to a period of one month. These sparse ground truths will be used as Boundary Conditions as stated in section 2. They contain an amount of pixels corresponding to $5\%$ of the pixels available in complete ground truths.

For the CAMS dataset, one sample is generated every three hours. Samples take the form of matrices of size 241x480. Depending on the chosen number of input modalities, each model input can be composed of one to seven of these matrices. The training set therefore contains 2680 of these inputs, the sparse training set (with sparse ground truths) 240 inputs, and the test set 2920 inputs.

For the ALADIN datasets, one sample is generated every hour. The size of the matrices is 405x613. Again, depending on the chosen number of input variables, each model input can be composed of one to seven of these matrices. The training set therefore contains 8040 of these inputs, the sparse training set (with sparse ground truths) 720 inputs. For the test set, we only use one image for every three hours, so that it contains as many samples as the CAMS test set. Therefore it also contains 2920 inputs.

An exhaustive study on our two models (UNets and GANs), three fusion strategies and six input variables (and all possible combinations of these) is conducted on the CAMS dataset. This corresponds to 192 different experiments. Only experiments that lead to the best performances on the CAMS dataset are conducted on the ALADIN dataset. The aim is to provides insight on the impact of the characteristics of each dataset on the results, as shown in section 6.2.

Then, experiments are realised on the hybrid approaches, but only on UNet models, and always with all six additional input variables. These experiments are realised on both datasets as well.

## 5.2 Data pre-processing

All values of AOD inferior to $0.005$ can be considered as noise. They are therefore set to 0 before being used, be it for training or prediction.

In order to speed up the convergence of the models, we equalise the PM2.5 and AOD distributions by applying the function $ln(1 + x)$ to those values. The inverse function $exp(x) - 1$ is then simply applied to inferred outputs in order to obtain actual concentration values (in $\mu g/m^3$ and ease the interpretation of our results.



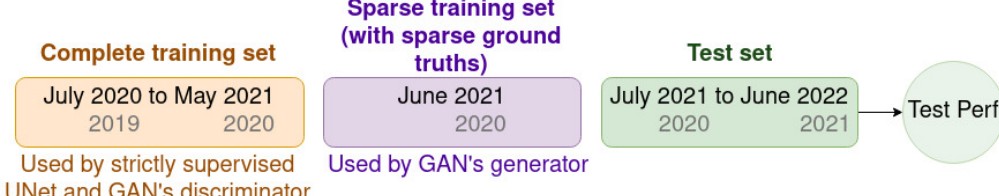

**Figure 7.** Representation of the datasets used for our experiments. The text in gray specifies the corresponding peridos of time for the datasets built with the data from the ALADIN model instead of the CAMS model.

In our context, polluted regions (with high aerosol concentration values) are our areas of main interest. Therefore filtering very low aerosol concentration values in our ground truths and predictions allows us to better evaluate the model performance
in these areas. Specifically, values inferior to $1\mu g/m^3$ are set to $0\mu g/m^3$. This is done before computation of the evaluation metrics.

These previous pre-processing protocols are based on the protocols proposed by (Dabrowski et al., 2023).

The data gives us access to values of both Eastward and Northward wind speeds. Instead of using them as such, we apply the transformation described by equation 1 to instead obtain two different matrices. In this equation, U and V respectively represent
the Eastward and Northward wind speeds. The first one contains values of the wind speed norm (regardless of direction), and the second one values of the direction (in degrees) of the wind. Each time wind values are used in an experiment, both of these matrices are used.

$$
\begin{aligned}
norm =&\sqrt{U^2 + V^2}\\
direction =&arctan\left(\frac{V}{U}\right).\left(\frac{180}{\pi}\right)
\end{aligned}
\tag{1}
$$

Our data does not originally contain values of the Angström exponent, but it is easily possible to compute them using values
from the AOD measured at two different wavelengths, and the equation 2. In this equation, $\lambda_1$ represents the wavelength of the original AOD, which is $550nm$, and $\lambda_2$ is the wavelength of the second AOD, used simply to compute the Ångström exponent. When using data from the CAMS model, this second wavelength is of $865nm$, while with the ALADIN model, it is of $1000nm$.

$$
angstrom(\lambda_1,\lambda_2) = -\frac{ln\left(\frac{AOD(\lambda_2)}{AOD(\lambda_1)}\right)}{ln\left(\frac{\lambda_2}{\lambda_1}\right)}
\tag{2}
$$

Pressure values are converted from Pascal to atmospheres, and temperature values from Kelvin to Celsius degrees.

### 5.3   Metrics and Losses

During training, depending on the type of model, different losses can be used. For our GANs, the adverse loss is used (to train both the generator and the discriminator), as well as the Boundary Condition loss (which is essentially a localised MSE as





stated in section 2). For our supervised UNets, we use the MSE loss function as well as the FSIM (Zhang et al., 2011), which
is also used as a metric and described in this section.

Four metrics are used to evaluate the models during testing: the Mean Absolute Error (MAE), the Mean Bias Error (MBE),
and the Quantized Error (QE) as proposed by (Dabrowski et al., 2023), as well as a metric proposed by (Zhang et al., 2011)
called the FSIM. The MAE and MBE are expressed in $\mu g/m^3$ (the unit of aerosol concentration) and in $\%$ (for their relative
versions).

In the following equations, $y$ represents the ground truth, $y_i$ its elements, and $\overline{y}$ its average. $\hat{y}$ is the output of the model and
$\hat{y}_i$ its elements. The number of elements in either matrix is $N$.

– **Mean Absolute Error (MAE)**

It is the most widely used of these metrics, it represents the error of the models in a general sense (Wang and Lu, 2018).

– **Mean Bias Error (MBE)**

It represents the model's tendency to overestimate or underestimate the values to predict. A model with poor performance
can however still have a low bias, as it is not the only aspect of its performance. This error can sometimes be used to
correct the bias of the model.

– **Quantized Error (QE)**

This metric is used to quantize the prediction as well as the ground truth before comparing them. The quartiles of the
ground truth values distribution are used to define four classes for this quantization.

$$C_{i,j} = \begin{cases} 0 & \text{if } M_{i,j} \leq q1 \\ 1 & \text{if } M_{i,j} \in ]q1,q2] \\ 2 & \text{if } M_{i,j} \in ]q2,q3] \\ 3 & \text{if } M_{i,j} > q3 \end{cases} \tag{3}$$

This process allows us to obtain a quantized ground truth $C_{gt}$ and a quantized prediction $C_{pred}$. The Quantized Error is
computed from these matrices following the equation (4).

$$QE = \frac{|C_{gt} - C_{pred}|}{N} \tag{4}$$

This type of metric is usually better suited for classification or segmentation tasks. However, the air quality is often
represented as indexed values, and involves thresholds corresponding to different levels of health hazards (or policy
alerts): this metric is therefore more closely related to this representation. It is also more sensitive than the MAE to very
localised errors. It is on the other hand less suited to represent tendencies to generally overestimate or underestimate the
values to predict than the MBE.





– **Feature Similarity Index (FSIM)**

This metric has been proposed by the authors of (Zhang et al., 2011), as an Image Quality Assessment metric. It relies on the concepts of Phase Congruency and Gradient Magnitude (in the sense of Image Gradient). The Phase Congruency is also used to weigh the contribution of each pixel to the similarity of two images. This leads to a significant weight being given to edges, shapes and other structures in the images.

**5.4 Scores**

Since a lot of different metrics are used, a protocol is needed to ease the comparison between models or experiences. Three different scores, computed from the previously described metrics, are used.

  – **Total score**

This score is computed using all of our metrics at the same time, including the inference time. It follows equation 5.
$rMAE$ and $rMBE$ represent the relative counterparts of $MAE$ and $MBE$, and are expressed in $\%$ instead of $\mu g/m^3$.

$$Total\ score = \frac{\frac{0.05s - Inference\ time}{0.05s} + (1 - rMAE) + \frac{3 - QE}{3} + (1 - rMBE) + (1 - FSIM)}{5} \tag{5}$$

  – **Timeless score**

This score is very similar to the first one, but does not include the inference time metric. This allows to identify the best performing models for a situation in which the inference time is not a predominant factor. It follows equation 6.

$$Timeless\ score = \frac{(1 - rMAE) + \frac{3 - QE}{3} + (1 - rMBE) + (1 - FSIM)}{4} \tag{6}$$

  – **Reduced score**

This score is computed using only $rMAE$ and $FSIM$. These two metrics are the most relevant ones in the Computer Vision domain. This score therefore allows for a comparison of the models and experiments from this point of view only. It also does not make use of the inference time. It follows equation 7.

$$Reduced\ score = \frac{(1 - rMAE) + (1 - FSIM)}{2} \tag{7}$$

**5.5 Methodology and data summary**

In order to ease the comparison of this work to other models using a scaling approach for inference of PM2.5 from AOD, we provide hereafter in Table 1 a summary of our methodology, as well as some characteristics of the dataset and methods used in this paper. It follows the standard proposed by the authors of (Zhou et al., 2024).



**Table 1.** Characteristics of the dataset, method and experiments used in this paper, in the standard proposed by the authors of (Zhou et al., 2024).

| Standard | Indicator | Description |
|---|---|---|
| **Dataset** | **Open source** | Yes. For more info, see Code and data availability section, just after section 7. |
| | **Data feature** | |
| | Predict step | Single step. |
| | Time resolution | Every 3 hours for CAMS, every hour for ALADIN. |
| | Data size | CAMS : 5840 samples. ALADIN : 11 680 samples. |
| | **Data dimensions** | Up to eight matrices per sample. Shape for CAMS : 241x480. For ALADIN : 405x613. |
| | **Dataset split** | Training set and test set both span over a year. |
| | | Test set always contains 2920 samples out of the total. |
| | **Pre-processing** | |
| | Missing value | Handled by CAMS and ALADIN models. |
| | Conversions | Pressure : Pa to atm. Temperature : K to °C. |
| | Filtering | AOD with a threshold of 0.005. |
| | Normalizing | Function $ln(1+x)$ applied to AOD and PM2.5 |
| | Others | Computing the Ångström Exponent out of AOD values. |
| | | Extracting the wind speed norm and direction from its northward and eastward speeds. |
| **Method** | **Open source** | Yes. For more info, see Code and data avilability section, just after section 7. |
| | **Architecture** | UNet, inspired from (Ronneberger et al., 2015). GAN, inspired from (Goodfellow et al., 2020). |
| | | Implementation of Data Fusion, Feature Fusion and Decision Fusion methods. |
| | **Training process** | Optimizer : Adam. Loss functions : MSE and FSIM (UNet), adversarial and based on BC (GAN). |
| | **Visual analysis** | Available in figures 11 and 13. |
| | **Novelty** | Implementation of a Hybrid Fusion method. |
| | | Study of the impact of several meteorological variables on the results. |
| **Experiments** | **Experimental setting** | |
| | Model config | Encoder kernel sizes : 9, 7, 7, 3. Decoder kernel sizes : 3, 7, 7, 9. Size of latent vector : 128. |
| | Computation setup | GPU Nvidia Tesla A100 with 80Go of V-RAM. |
| | **Results metrics** | MAE, MBE, FSIM and QE (non-classical, see section 5.3). |
| | **Modeling metrics** | |
| | Params | Depending on models, between 10 and 250 millions of trainable paramters. See figure C1. |
| | **Comparison with SOTAs** | Model outperforms the kriging method, Polynomial Regression of Degree 3, |
| | | and Random Forest algorithms. |
| | **Ablation study** | Yes. Several models with several information fusion methods are tested. |
| | | Experiments are realised with different numbers of meteorological variables as input. |





## 6  Results

We start with a general overview of our results. The next step is to identify the best performing and most interesting results and models among the experiments realised on the CAMS dataset. This allows us to reproduce these experiments on the ALADIN dataset, in order to compare the results and better understand the impact of the characteristics of these datasets (namely, spatial domain and resolution) on the results.

### 6.1  Overview

We choose to summarize our results in the form of radar charts, with five metrics represented on these charts: the inference time $t$, $rMAE$, $QE$, $rMBE$ and $FSIM$. For simplicity, Figure 8 displays the common legend for all these charts.

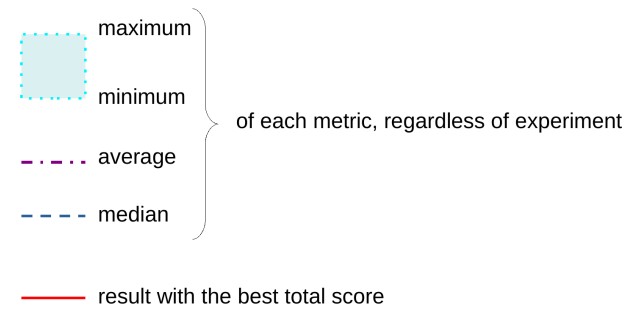

**Figure 8.** Common legend for all radar charts.

Figure 9 gives an overview of the performance of each couple model-fusion strategy.

It shows that, on average, models using Decision Fusion have the longest inference time, while models using Data Fusion are the fastest and those using Feature Fusion are in the middle. This is expected, as it correlates with the number of parameters of each model, as illustrated by figure C1.

It also shows that GANs seem to generally suffer from poorer $rMAE$ and $rMBE$ scores than UNets. Our interpretation is that the proportion of the training set reserved for strictly supervised training is important enough for purely supervised methods to perform well. The interest of GANs lies in their ability to realise semi-supervised training. In our case, it also corresponds to their ability to make use of the portion of our dataset that only contains sparse ground truths. This portion is small, which therefore makes the interest of GANs (and arguably semi-supervised methods) limited in this case. In comparison, the authors of (Dabrowski et al., 2023) show the efficiency of their GANs in a context where only half of the dataset contains complete ground truths.

It is interesting to note that the $FSIM$ and $QE$ metrics do not seem to be affected by this in the same way, or at least not as intensely.

Finally, the Data Fusion or Channel Concatenation (CC on the figure) strategy seems to be leading to more stable results than the other two fusion strategies. This may also be linked to the difference in model complexity, as shown in section C.



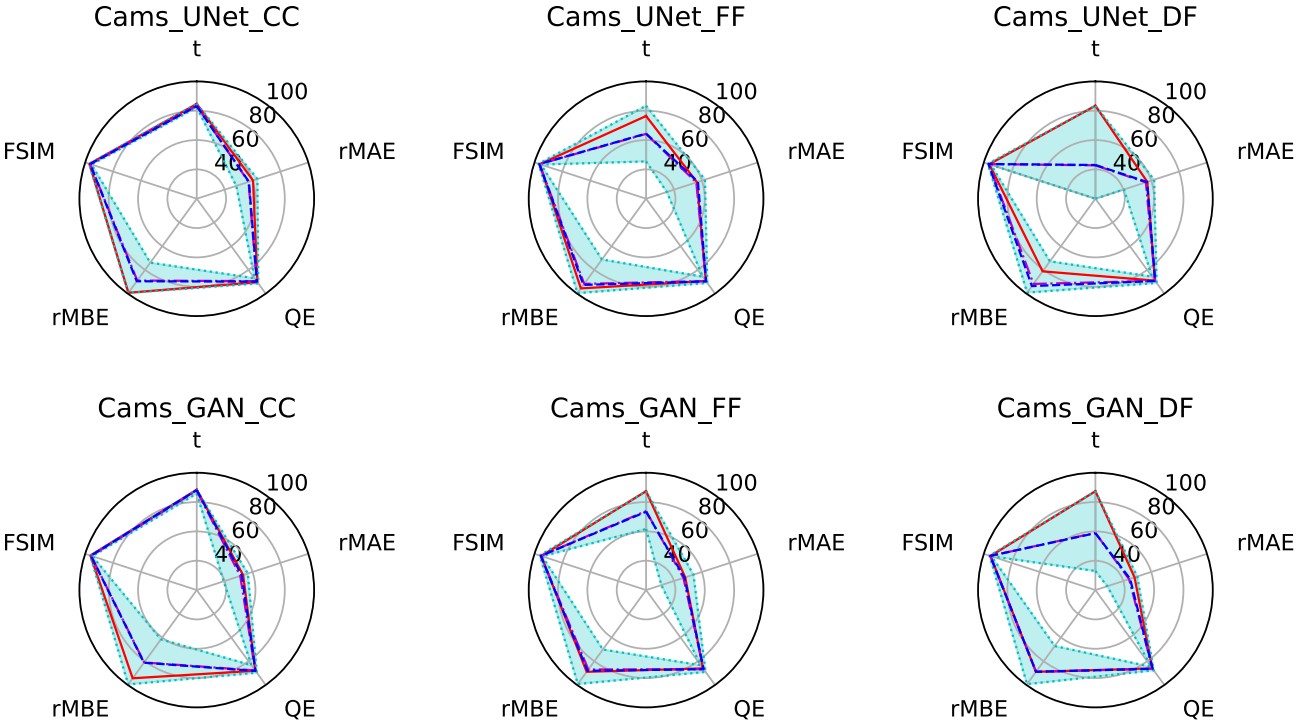

**Figure 9.** Overview of each couple model - fusion strategy. The charts names shows information about the chosen dataset, the model, and the fusion type. Regarding the fusion type, 'CC' means 'Channel Concatenation' or Data Fusion, 'FF' means Feature Fusion, and 'DF' means Decision Fusion.

Figure 10 shows the evolution of the performances of our UNet when we increase the number of input variables.

This Figure shows that, when increasing the number of input variables, the inference time lowers. This is expected, as figure
440 C1 shows that the the models grow in complexity with the number of input variables. Other metrics, and especially $rMAE$, show on average an increase in performance when adding more input variables.

However, this increase in performance is not linear, and when deciding to add an input variable to a given experiment, we are not guaranteed to obtain better results. We can also note that, when comparing experiments with two additional input variables to three, we observe less stable $rMBE$ values, even though the number of experiments for these two categories is the same.
445 Finally, these charts also show that, regardless of the fusion method used, experiments realised using all five input variables tend to produce the best results (apart from the point of view of the inference time).





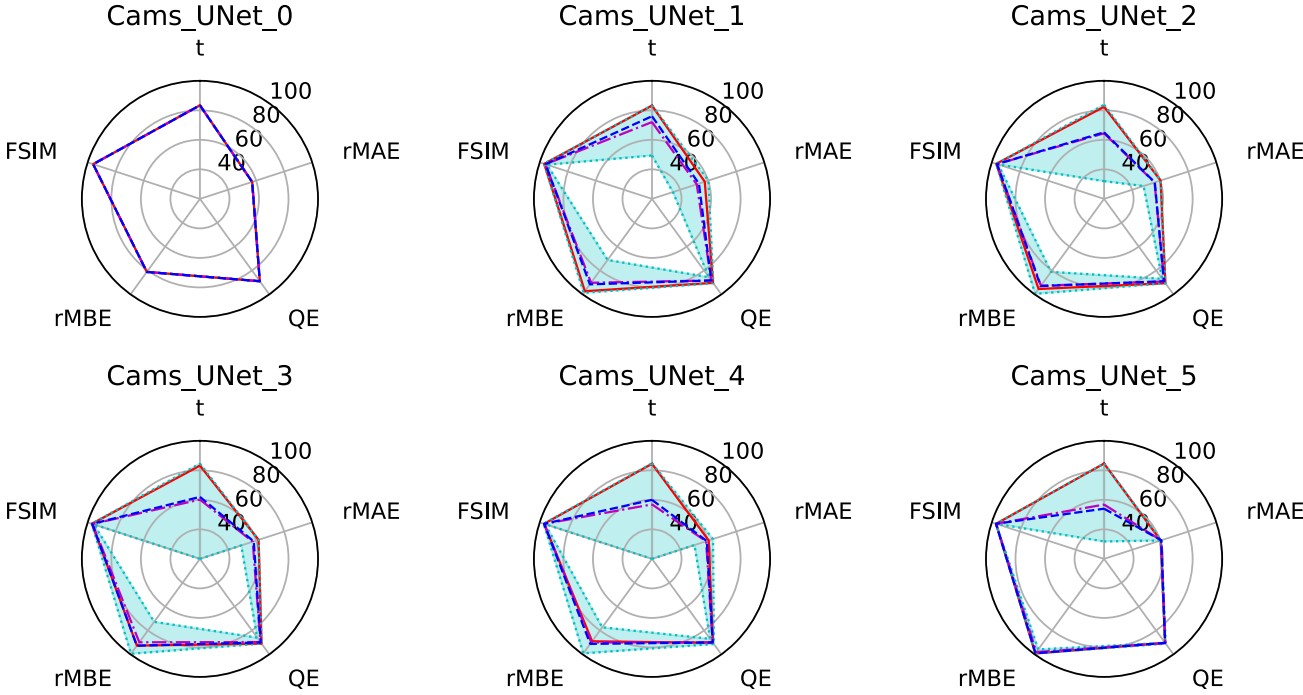

**Figure 10.** Overview of the evolution of our UNet's performances when increasing the number of input variables. The wind is counted as one variables even though it contains two channels (one for wind speed and a second one for direction).

## 6.2 Best results on the CAMS dataset

Table 2 shows the best results according to each of the three scores. First line is the result with the best Total Score, second line is the best Timeless Score, and third line is the best Reduced Score. Figure 11 shows the output of these models for one given sample.

This table shows that using more variables as input seems to generally lead to the best results, except for the Ångström Exponent on the last line. It also shows that Decision Fusion methods suffer from substantially larger inference times, especially compared to Data Fusion.

These results lead to three main recommendations depending on the context and the desired performances. If the $MBE$ (or bias) of the output is not an important factor, then the recommended model is a UNet using the Data Fusion strategy, as well all proposed input variables, except for the Ångström exponent. If the Inference Time is not an important factor, then the use of a UNet model with the Decision Fusion strategy and all proposed input variables is advised. Finally, a UNet with the Data Fusion strategy as well as all proposed variables gives the most balanced results.



**Table 2.** Best results on CAMS Dataset. In the column "variables used", we put the initial of each used variable: W for Wind, H for Humidity, P for Pressure, T for Temperature, A for Ångström Exponent. The AOD is always used as input. The column "fusion type" contains "Data" for Data Fusion, "Feature" for Feature Fusion, "Decision" for Decision Fusion. The symbol ($\nearrow$) means the value(s) is to be maximized: a high value means a good performance. If the symbol is absent, then the concerned value is to be minimized (a low value means a good performance).

| Model Type | Fusion Type | Variables Used | Inference Time | MAE | QE | MBE | FSIM | Scores ($\nearrow$) | | |
| --- | --- | --- | --- | --- | --- | --- | --- | --- | --- | --- |
| | | | | | | | | Total | Timeless | Reduced |
| UNet | Data | WHPTA | **0.0077** | 4.38 | 0.29 | -0.13 | 3.10% | **86.28%** | 86.7% | 78.61% |
| UNet | Decision | WHPTA | 0.0341 | 4.33 | 0.29 | **0.05** | 2.95% | 75.8% | **86.81%** | 78.93% |
| UNet | Data | WHPT | 0.0086 | **4.04** | **0.26** | -2 | **2.8%** | 83.38% | 83.52% | **80.33%** |

Looking at the left column Figure 11 gives us a bit more insight into the results. On this sample, it seems that the model using Data Fusion and all variables except the Ångström exponent is the one providing the best prediction for the area n°2 on the image. Other models overestimate the aerosol concentration in this area. The fact that this model has the worst $MBE$, and that it is negative could show a tendency to underestimation. The observations made on this sample are coherent with this assumption.

The model using Data Fusion and all variables however provides good estimations for these two areas. Finally, the model using Decision Fusion (and all variables) underestimates the concentration in area n°1 and overestimates it in areas n°2 and 3. This model has the best $MBE$, but not the best $MAE$. Our hypothesis is that it overestimates certain areas and underestimates others, which compensates and leads to a small bias.

### 6.3   Comparison between the CAMS and ALADIN datasets

Table 3 shows the results obtained for the same models as in section 6.2 but on the ALADIN dataset, and the right column of Figure 11 shows their outputs for one given sample.

**Table 3.** Results on ALADIN Dataset. In the column "variables used", we put the initial of each used variable: W for Wind, H for Humidity, P for Pressure, T for Temperature, A for Ångström exponent. The AOD is always used as input. The column "fusion type" contains "Data" for Data Fusion, "Feature" for Feature Fusion, "Decision" for Decision Fusion. The symbol ($\nearrow$) means the value(s) is to be maximized: a high value means a good performance. If the symbol is absent, then the concerned value is to be minimized (a low value means a good performance).

| Model Type | Fusion Type | Variables Used | Inference Time | MAE | QE | MBE | FSIM | Scores ($\nearrow$) | | |
| --- | --- | --- | --- | --- | --- | --- | --- | --- | --- | --- |
| | | | | | | | | Total | Timeless | Reduced |
| UNet | Data | WHPTA | **0.0064** | 8.37 | 0.35 | -3.92 | 4.37% | 84.03% | 83.24% | **80.02%** |
| UNet | Decision | WHPTA | 0.0364 | **8.29** | **0.34** | 1.65 | **3.97%** | 73.41% | 84.96% | 79.99% |
| UNet | Data | WHPT | 0.0075 | 8.76 | 0.37 | **-1.0**4 | 4.64% | **85.39%** | **85.49%** | 78.79% |





**Figure 11.** Outputs of the best models on both datasets for one given sample. The pink circles and numbers have been added afterwards to attract the reader's attention on some details of the image.




This table shows an important difference between the results obtained on the CAMS and the ALADIN dataset. However, even though the same metrics are used, these sets of results are not easily comparable to each other, as they are obtained on different data. Indeed, the ALADIN dataset contains images of much higher resolution than the CAMS dataset, these images do not represent the same geographical domain (Europe for ALADIN, the world for CAMS), and these dataset do not correspond to the same time period (July 2020 to June 2022 for CAMS, July 2019 to June 2021 for ALADIN). This explains why, in the CAMS dataset, the aerosol concentration values are comprised between 0 and $34\,425\ \mu g/m^3$ with an average of $11.02\ \mu g/m^3$, while in the ALADIN dataset, they are comprised between 0 and $6\,774\ \mu g/m^3$ with an average of $23.17\ \mu g/m^3$.

Figure 11 also shows a difference between results on the CAMS and ALADIN datasets. The model using Data Fusion with all variables underestimates the aerosol concentration in areas n°1 and 3, which is consistent with the fact that this model has the lowest $MBE$ out of the three. The model using Data Fusion and all variables except the Ångström exponent also underestimates concentration in these areas, but less so. This is coherent with the fact that of all three models, this one has the second lowest $MBE$. The model using Decision Fusion does not underestimate concentration in areas n°1, underestimates the concentration in area n°3 as all other models, and overestimates the concentration in area n°2. It is also the model with the highest $MBE$ and the best $MAE$ out of all three.

Comparison between results on our two datasets does remain interesting, as the best performing methods on the CAMS dataset do not seem to correspond to the best performing ones on the ALADIN dataset.

For example, let us look at the results from the table obtained with the Data Fusion strategy. One of these results is obtained while using all available variables as input, and the other is obtained using all variables except the Ångström exponent. Based solely on these two results, on the CAMS dataset it would seem using the Ånsgtröm exponent as part of the input variables leads to a smaller MBE, but we obtain higher values for all other metrics (except the inference time). On the ALADIN dataset the same situation and decision (of using the Ångström exponent) seem to lead to opposite results (higher MBE, smaller rest of the metrics).

This shows that the impact of the use of one specific input variable on the results of our models can not easily be interpreted. This is due to the interaction between the input variables themselves, and the very nature of the Neural Networks, which are often described (with reason) as black boxes.

## 6.4 Interpretation of the impact of the Ångström exponent

Let us look at Figure 12 to try and understand the impact of using the Ångström exponent on our results on the CAMS Dataset. This figure shows that the two metrics that are impacted the most by the use of the Ångström exponent are the $MAE$ and the $MBE$ (and their relative counterparts). Using the Ångström exponent seems to lead to a higher minimum value for the $MAE$. In other words, it helps avoiding our worst results (w.r.t. the $MAE$). The best $MBE$ values are obtained when using the Ånsgtröm exponent. Using it therefore seems to lead to a lower bias.

Once again, these observations are valid for the CAMS Dataset, and for the chosen periods. We can not make a general conclusion on the use of the Ångström exponent as an input variable based on these observations alone. In particular, these observations are consistent with the results shown in table 2 (obtained with the CAMS Dataset), but not with those in table 3



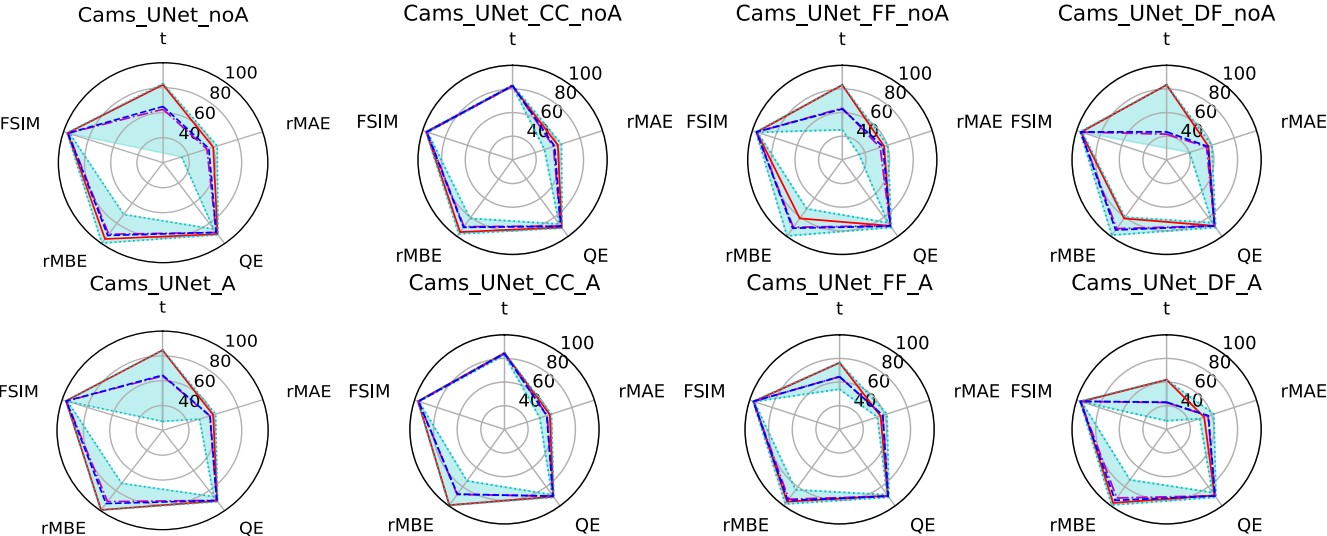

**Figure 12.** Overview of our experiments with (first line) and without (second line) using the Ångström exponent as an input variable. The charts names show information about the chosen dataset, the model, and the fusion type. Regarding the fusion type, 'CC' means 'Channel Concatenation' or Data Fusion, 'FF' means Feature Fusion, and 'DF' means Decision Fusion. The first column corresponds to all experiments regardless of the fusion strategy used.

505   (obtained with the ALADIN Dataset). This shows that our observations (about the Ångström exponent) on the CAMS Dataset can not automatically be assumed to be true for the ALADIN Dataset too.

## 6.5   Results of hybrid fusion method

Tables 4 and 5 shows the results of the two hybrid models described in section 4.3.4 on the CAMS and ALADIN datasets, respectively. Figure 13 shows the outputs of these models on one given sample. These results show that, from an Artificial

**Table 4.** Results of hybrid models on the CAMS Dataset. The column "fusion type" contains "Hybrid1" for the model represented by Figure 5 and "Hybrid2" for the model represented by Figure 6. The symbol ($\nearrow$) means the value(s) is to be maximized: a high value means a good performance. If the symbol is absent, then the concerned value is to be minimized (a low value means a good performance).

| Model Type | Fusion Type | Inference Time | MAE | QE | MBE | FSIM | Scores ($\nearrow$) | | |
|---|---|---|---|---|---|---|---|---|---|
| | | | | | | | Total | Timeless | Reduced |
| UNet | Hybrid1 | **0.0098** | 4.39 | 0.28 | **0.2** | 2.96% | **85.25%** | **86.46%** | 78.67% |
| UNet | Hybrid2 | 0.0116 | **4.1** | **0.27** | -0.86 | **2.9%** | 84.09% | 85.91% | **80.03%** |

510   Vision point of view, the second proposed hybrid model is better. However the first model appears to be more balanced, and is recommended in any situation where the $MBE$ and Inference time are important metrics.





These models, while showing satisfying performance, show poorer performances than some of the results presented in table 2. Therefore we do not recommend the use of these hybrid models with the CAMS Dataset.



**Figure 13.** Outputs of hybrid models on both datasets for one given sample. The pink circles and numbers have been added afterwards to attract the reader's attention on some details of the image.



The left column of Figure 13 shows that both hybrid models produce a relatively adequate estimation for area n°3, under-
estimate concentration in area n°1, and overestimate it in area n°2. This is coherent with both models having relatively close
metric values, and having $MBE$ values close to $0$. These results show that, on the ALADIN dataset, the first proposed Hybrid

**Table 5.** Results of hybrid models on the ALADIN Dataset. The column "fusion type" contains "Hybrid1" for the model represented by
Figure 5 and "Hybrid2" for the model represented by Figure 6. The symbol ($\nearrow$) means the value(s) is to be maximized: a high value means
a good performance. If the symbol is absent, then the concerned value is to be minimized (a low value means a good performance).

| Model Type | Fusion Type | Inference Time | MAE | QE | MBE | FSIM | Scores ($\nearrow$) | | |
|---|---|---|---|---|---|---|---|---|---|
| | | | | | | | Total | Timeless | Reduced |
| UNet | Hybrid1 | 0.0108 | **7.69** | **0.33** | **-1.42** | **4%** | **85.21%** | **86.91%** | **81.52%** |
| UNet | Hybrid2 | **0.0103** | 7.9 | **0.33** | -2.87 | 4.2% | 83.91% | 85.04% | 81.05% |

model leads to better results than the second, on all metrics (except the Inference time).

The results obtained with this model are also better than all results presented in table 3. However, the model that were tested
on the ALADIN dataset only correspond to the model that produced the best performances on the CAMS dataset. This means
that we can not conclude from these results that the Hybrid models work better than other models on the ALADIN dataset. To
arrive to such a conclusion, we would need to realise an exhaustive study on our three fusion strategies, two models (GAN and
UNet) and six input variables.

The right column of Figure 13 shows that the first hybrid model slightly underestimates concentration in areas n°1, and the
second hybrid model underestimates it more. The first model slightly overestimates concentration in area n°2, while the second
model provides a more accurate estimation. These observation are coherent with both models having a low $MBE$ and the
second model having the lowest of the two. Interestingly enough, both these hybrid models seem to propose a better estimation
of area n°3 than the models shown in Figure 11.

### 6.6 Comparison with SOTA

Table 6 shows a comparison of our best results with a few methods used as baseline on the CAMS dataset. The total score is
not shown for the Random Forest algorithm and the kriging method, because of their high inference times. It can however be
considered as being significantly worse than the total score of any other discussed method, for the same reason.

It is important to note that the Polynomial Interpolation and Random Forest Algorithm only use the AOD as input, while the
kriging method only uses sparse values of the aerosol concentration (which represent our Boundary Conditions) as input.

The Polynomial Interpolation method has a significantly smaller inference time than any other method discussed in this
paper. However this is the only metric on which one of the baseline methods outperforms our best results. Indeed, our models
outperform the chosen baselines by a large margin, in all metrics except this one and all scores .





**Table 6.** Comparison of baseline models with our best results on the CAMS dataset. Our best results on the CAMS dataset are those presented in table 2. "Poly. Interp." stands for Polynomial Interpolation, "Ord." for Ordinary, and "HE" for "hole-effect". The symbol (↗) means the value(s) is to be maximized: a high value means a good performance. If the symbol is absent, then the concerned value is to be minimized (a low value means a good performance).

| Model Type | Fusion Type | Variables Used | Inference Time | MAE | QE | MBE | FSIM | Scores (↗) | | |
|---|---|---|---|---|---|---|---|---|---|---|
| | | | | | | | | Total | Timeless | Reduced |
| Poly. Interp. of Degree 3 | AOD only | **0.0007** | 6 | 0.41 | -3 | 4.96% | 79.62% | 74.87% | 70.34% | |
| Random Forest Algorithm | AOD only | 1.1535 | 6.01 | 0.41 | -2.63 | 4.95% | *NA* | 75.7% | 70.33% | |
| Ord. kriging with HE variogram | BC only | 40.3732 | 6.03 | 0.38 | -2.99 | 7.67% | *NA* | 78.01% | 73.51% | |
| UNet | Data | WHPTA | 0.0077 | 4.38 | 0.29 | -0.13 | 3.10% | **86.28%** | 86.7% | 78.61% |
| UNet | Decision | WHPTA | 0.0341 | 4.33 | 0.29 | **0.05** | 2.95% | 75.8% | **86.81%** | 78.93% |
| UNet | Data | WHPT | 0.0086 | **4.04** | **0.26** | -2 | **2.8%** | 83.38% | 83.52% | **80.33%** |

Our hybrid models do not appear in table 6, as they are outperformed by the models presented in this table. However, as stated in section 6.5, their performances remain comparable. In other words, our hybrid models are outperformed by the models presented in table 6, but not by a large margin.

It has already been stated that, in a context where ground truths were less available, GANs would outperform UNets. Indeed their usefulness for this problem lies in their ability to realise semi-supervised learning. It is also interesting to note that generally speaking, all models would probably benefit from a larger amount of data, as long as the training set remains representative of the actual data in a real-case scenario. The representativity of the dataset is paramount as it helps avoiding the overfitting problem often encountered in machine learning. More specifically, our deep learning models are the ones that would benefit the most from a larger amount of accessible data, as they contain more parameters.

# 7 Conclusion

In this paper we performed an extensive study on the use of several meteorological variables and column aerosol optical properties as inputs for a Deep Learning model to infer PM2.5 concentration from AOD using a scaling approach applicable globally. We tested different network architectures as well as the use of three different fusion strategies for the exploitation of these inputs, in order to investigate the optimal way of fusing those information for our specific application. Hybrid methods of fusion have been proposed, implemented and studied as well. Our experiments were conducted extensively on CAMS data in order to assess model performances at global scale. We also performed limited experiment using the ALADIN dataset (instead of CAMS) over a large region covering Europe and the Mediterranean basin to study the impact of the datasets' characteristics on our results, especially its spatial resolution and geographic spatial coverage.

Based on five metrics used throughout to evaluate different models performances, our experiments have shown the superiority of UNets over BC-GANs in our context, as is shown by figure 9. However the sparse training set is, in our context,



significantly smaller than the complete set. We suggest in section 6.1 that this induces a reduced need for semi-supervised learning, and explains the difference in performance between UNets and GANs. The authors of (Dabrowski et al., 2023) show the superiority of their BC-GANs over UNets in their context, which includes sparse and complete training sets of more comparable size. This shows that the difference in performance between BC-GANs and UNets is not inherent to these models themselves. Therefore we recommend the use of a UNet in our context of semi-supervised learning, with our sparse training set being significantly smaller than our complete training set. It remains difficult to deduce a superiority of one model over the other in the general sense from our experiments. The comparison between our results and the results of (Dabrowski et al., 2023) does show that the quantity of sparse data has a significant impact on the performances of these two models. Therefore this context parameter must be taken into account when recommending one of these models over the other.

Our results have also illustrated that increasing the number variables as input tends to augment model performances. This is not surprising as the limited set of variables we used were selected for their known influence on PM2.5 surface concentration. This remains a tendency however, and not a guarantee as some exceptions have been observed where, depending on network architecture and fusion strategy, adding a variable may degrade performance. This is interesting because it is counter-intuitive to the general belief that using more (relevant) data in deep learning yields better results and emphasises in particular the interest of studying the impact of network architecture for atmospheric applications.

Our experiments have also shown in section 6.3 the importance of dataset's characteristics (here spatial resolution and coverage) and its impact not only on the results but on the conclusions that can be drawn from them as well. This is especially important in atmospheric sciences because geophysical variables have different scales of variability and network architecture should ideally be aligned with the spatial characteristics of input fields. Our work suggests that more work is needed to understand the impact of networks architecture on their ability to fully capture spatial features that are specifics to atmospheric sciences.

While identifying precisely the impact of each variable on the models' performances would be useful, the observations made in section 6.4, and drawn from our results, highlight the difficulty of such a task.

The two fusion strategies that lead to our best results (shown in section 6.2) are the Data and Decision Fusion ones. According to our experiments, the Data Fusion strategy also seems to lead to more stable results. Moreover, it allows to build smaller models, which in turn leads to shorter inference times and training times.

Our experiments on hybrid models did not show clear evidence of their advantage compared to other models, even though they do present comparable performances, as shown in section 6.5. Based on these conclusions, the Data Fusion strategy is the one we would recommend in a general case when all input variables are available at the same resolution and over the same area. Of course this recommendation depends on general context and more specifically on the definition of the desired outcome. For example, using different metrics to measure performance might lead to a different recommendation.

Finally, our objective was not to develop a single and optimized model for PM2.5 inference from AOD but rather to study how multiple PM2.5 predictors could be used in order to best align the network architecture with the seek inference function. However, and while we did not try to conduct specific optimization and used only a limited set of predictors, we have proposed several architectures that yield PM2.5 inference performances comparable to other tailored models found in the literature (Ma



et al., 2022; Unik et al., 2023). The demonstrated performances obtained here should only be interpreted as baseline capabilities of the proposed models that could most likely be improved by extending further the time coverage of the learning database. As suggested by (Zhou et al., 2024), we also strongly encourage a more systematic evaluation of models against a common test dataset and using standardized metrics. Since the code and data used for this article are both available, we suggest that our current results be seen as a benchmark for the task and context presented in this paper. Such a benchmark could be used as a common ground for the evaluation of newly developed models of PM2.5 inference AOD data, therefore facilitating their comparison.

As stated before, the experiments realised in this paper have clearly illustrated the interest of using additional, carefully chosen, input variables in order to augment the performance of a scaling model to infer PM2.5 from AOD. Here we selected a limited number of meteorological variables and optical properties that are well known to drive surface level PM2.5 concentration. These variables are typically useful to establish the link between PM2.5 and AOD through a purely physics based model and not surprisingly our results demonstrate they are useful to establish this link through an Artificial Neural Network. Based on this insight, an interesting possibility of future work consists in applying the concept of Physics-Informed Neural Networks (described in detail in section A) to this problem and study, depending of the fusion strategy used, at which level the incorporation of physics equations would be most relevant.

*Code and data availability.*

– The code used for these experiments is available on a Zenodo archive (Dabrowski, 2024a).

https://doi.org/10.5281/zenodo.13947256

The data from the CAMS model used during these same experiments is available on a different Zenodo archive (Dabrowski, 2024b).

https://doi.org/10.5281/zenodo.13929498

– The data from the ALADIN model was extracted from the dataset proposed by (Mallet and Nabat, 2024).

https://doi.org/10.25326/703



**Appendix A: Related works**

**Generative Adversarial Networks (GANs).** Since the authors of (Goodfellow et al., 2020) proposed this type of model, the
popularity of GANs has increased consistently. They rely on the training of two networks : a generator and a discriminator.
The discriminator is presented with samples which can be either taken from the original data distribution, or generated (by the
generator). Its main task is to differentiate these two kinds of inputs. On the other hand, if the discriminator makes an error and
classifies a generated sample as real, then the generator is getting closer to its goal. The discriminator and generator's losses are
built in such a way that when one increases, the other decreases, and reversely. This why they are called adversarial networks.

Convolutional GANs are known for their ability to produce realistic images, which can fool both their discriminator but
also in some cases humans. They have also shown interesting performance in Image-to-Image translation tasks (Wang et al.,
2019, 2020; Zhu et al., 2017b). This type of task is usually categorized as paired image translation such as in (Isola et al.,
2017), or unpaired image translation such as in (Zhu et al., 2017a). In this article, our image translation task is a paired one.

**Explainable Artificial Intelligence (XAI).** Even though this work can not be classified as belonging to the field of XAI, the
terminology this field proposes remains interesting in the context of this work. The general idea behind XAI is to build models
that can be understood by their users, or whose results can. While this field has existed for several decades (Confalonieri
et al., 2021), its recent growth in popularity can be seen as a response to concerns about the black-box aspect of some neural
networks models. This growth has been particularly remarkable in applications fields like finance, medicine, law, and even
scientific production (Beckh et al., 2021; Murdoch et al., 2019; Belle and Papantonis, 2021; Roscher et al., 2020). In those
fields, the ability to explain a model and its results can represent the ability to ensure safety, fairness or scientific rigour. In a
more general sense, it makes it easier for the user to trust the model.

According to (Roscher et al., 2020), in the context of XAI, there are three important elements to consider when evaluating
the explainability of a model.

1. **Transparency:** an model is transparent if the processes that extract model parameters from training data and generate
   labels from testing data can be described and motivated by its designer.

2. **Interpretability:** its is the ability to generally understand what the model bases its decisions on. Some approaches for
   intepretable models are based on decision trees, as they can allow for an intuitive look on the decision-making process
   of a model.

3. **Explainability:** an explanation is the collection of features of the interpretable domain, that have contributed for a given
   example to produce a decision. For a model to be explainable, it generally needs to be possible to understand why the
   model's decision for a given datum A is different than for a given datum B.

It is interesting to note that domain knowledge can be used to enhance the explainability of a model (Beckh et al., 2021). In
this sense, Physics-Informed Neural Networks can be seen as a type of XAI.

**Physics-Informed Neural Networks (PINNs).** Physics-Informed learning, introduced by the authors of (Raissi et al., 2019),
can be considered today as its own research field. The term of Informed Networks suggest that the method makes use of prior



information about some specificity of the problem, for example its geometry. Physics-Informed Networks specifically make use of the physics of the problem to enhance their performances. This is usually done through the design of a physics-informed loss function, used during the training of the model. This loss function is often based off a differential equation that is verified by the data the model is using. As it can sometimes guide the training in a non-data oriented way, the use of this loss function reduces the need of these networks for labeled data, making them especially suitable for semi-supervised learning.

In physics, it is often necessary to use Initial and Boundary Conditions (BCs) to solve a given problem. In the literature around physics-based learning methods, two methods to take these BCs into account during training can be found. The soft constraint (or method) proposes to train the model to respect the BCs, through the use of an additional, tailored loss function. The hard constraint (or method) works through the transformation of the model outputs to enforce the respect of the BCs, and relies on pre-existing loss functions. When it comes to PINNs, the authors of (Sun et al., 2020) show that the hard constraint performs better than the soft.

Several authors have proposed to leverage the advantages shown by both adversarial and physics-informed approaches (Thanasutives et al., 2021; Nie et al., 2021), often calling these new models PI-GANs (Yang et al., 2019, 2020).

**Kriging method.** This spatial interpolation and extrapolation method was formalised by the author of (Matheron, 1963). In the statistical interpretation of the term, it is the optimal estimation method according to (Gratton, 2002). It is mathematically described by equation A1.

$$F(x_p) = \sum_{i=1}^{m} W_i \cdot F(x_i) \tag{A1}$$

$F(x_p)$, the value of function $F$ at point $x_p$, can be estimated thanks to $m$ surrounding points $x_i$, as the value of $F$ at these points is known. However it remains necessary to determine the weights $W_i$ of these points. The kriging method proposes to realise this through the estimation of what is called a variogram. To compute it, values of the variance of two points, and of the distance between them, are needed.

This method has been described as performing better when provided with a significant volume of data, and when the values to estimate are follwing a normal distribution.

For each inference, new points $x_i$ are used. As these points are the basis for the building of the kriging model, a new one is built for each inference. Because of this, kriging suffers from long inference time when compared to other methods presented in this article.

## Appendix B: Details of the approach

The purpose of this section is to show the architecture of the GAN's discriminator with our three main fusion strategies.

### B1 Data Fusion / Channel Concatenation

Figure B1 shows the GAN's discriminator's architecture with the data fusion strategy.Related





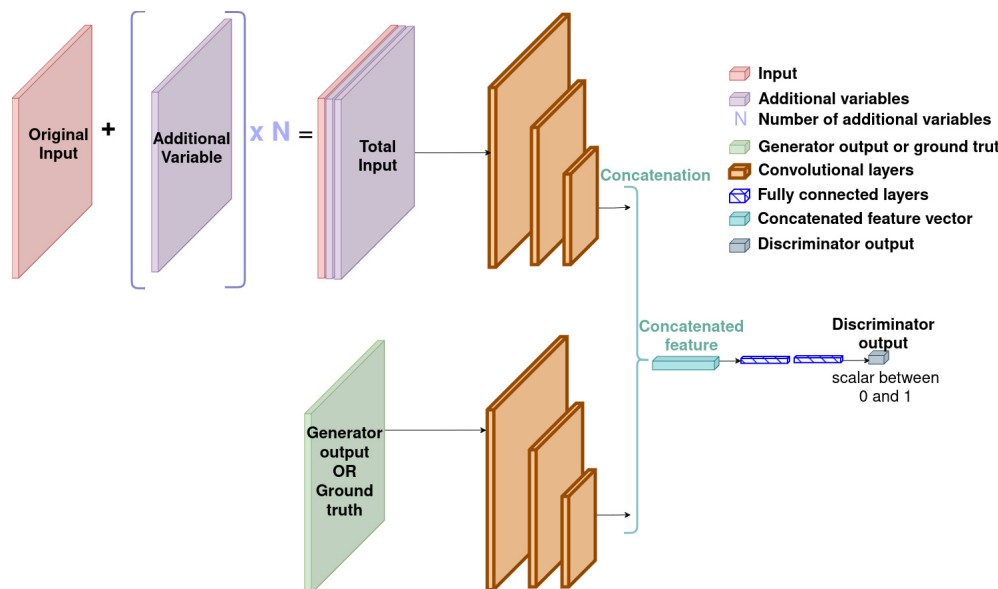

**Figure B1.** Architecture of the GAN's discriminator with data fusion approach.

## B2 Feature Fusion

Figure B2 shows the GAN's discriminator's architecture with the feature fusion strategy.

## B3 Decision Fusion

Figure B3 shows the GAN's discriminator's architecture with the data fusion strategy.

## Appendix C: Models complexity

Figure C1 shows the number of parameters of our models depending on the fusion method and the number of input images used. They correspond to the number of parameters for our UNets and BC-GANs, with both ALADIN and CAMS data.

*Author contributions.*

**Matthieu Dabrowski** : main author, designed the models and experiments, performed the experiments, led the analysis and the writing of this article.

**José Mennesson** : provided insights on the models and metrics used, helped in analysing the results, and contributed to the writing of this article.



**Figure B2.** Architecture of the GAN's discriminator with feature fusion approach.

**Jérôme Riedi** : defined the atmospheric model datasets to be used in the experiments, provided insights about physics of input variables
and PM2.5 values, helped in the analysis and the general writing of this article





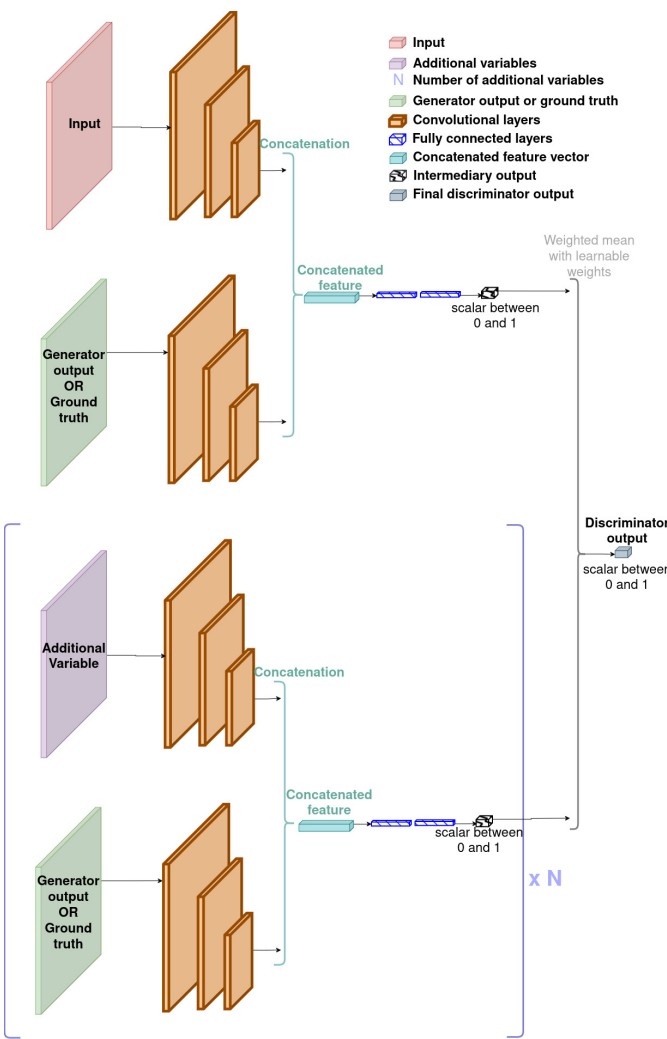

**Figure B3.** Architecture of the GAN's discriminator with decision fusion approach.

**Chaabane Djeraba** : provided insight on the NN models and on the general research strategy, as well as useful and welcomed tips for the writing of this article.

**Pierre Nabat** : provided the ALADIN data and insights into specifics of atmospheric composition models.

*Competing interests.* The authors declare an absence of competing interests.



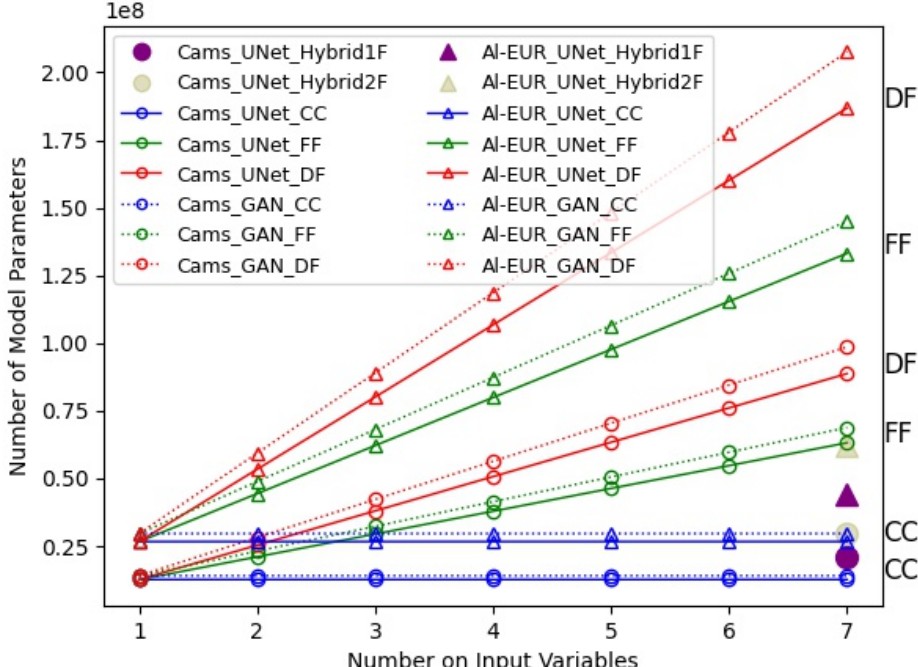

**Figure C1.** Number of parameters of each of our models depending on the number of input variables.

*Acknowledgements.* This work was partly supported by IRCICA USR 3380 (CNRS, Univ. Lille, F-59000 Lille, France), and has been made possible thanks to the financial support of several organizations, namely : the Agence Nationale de Recherche (ANR), the Centre Nationale d'Etudes Spatiales (CNES) and the Hauts-de-France region. Similarly, the University of Lille and the INRIA Center conjointly launched in 2019 the AI_PhD@Lille program, to which we owe the very existence of this project as well.





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
