# Peer review of "Knowledge-inspired fusion strategies for the inference of PM2.5 values with a Neural Network"

_EGUsphere, 2024_

## Author Response (AR1)

**Dear First Reviewer,**

First, we would like to thank the reviewer for their helpful comments and observations. We have taken into account most of the suggested modifications into a new version of this manuscript or are providing justifications otherwise.

1. *My main suggestion is the writing style in the results and conclusions sections (i.e., Sections 6 and 7). The descriptions were broken into multiple discontinuous small paragraphs, making the reading pretty hard. It reads more like a draft or oral presentation, instead of a research article. I would suggest reorganizing each subsection into a handful of coherent 'big' paragraphs.*

In our new version, section 6 and 7 have been reorganised into fewer and more self contained paragraphs.

1. *Line 25: (Martin et al., 2019) report –¿ Martin et al. (2019) report*
2. *Line 93: for PM2.5 which results and performances —¿ for PM2.5 whose results and performances*
3. *Line 125: caracterize –¿ characterize*

Each of these typos and formulations have been corrected the suggested way in the new version of our manuscript.

1. *Lines 319 and 362: Please formally describe the Boundary Conditions-GAN/loss (in a mathematical way).*
2. *Section 5.3: Please provide the mathematical equations for MAE, MBE, and FSIM*
3. *Eqs.(3) and (4): Please provide the definitions of $M_{i,j}$, $C_{i,j}$, and $N$*

Our new version now contains more detailed mathematical descriptions of almost all these elements. The only exception concerns the mathematical formulation of the FSIM metric.

Indeed, the FSIM is a complex image quality indicator, based on the concepts of Gradient Magnitude and Phase Congruency, which conception and computation details can not be easily nor briefly summarized within our own paper through a few simple equations. Instead several complex intermediary equations and mathematical concepts would need to be explained in details for a mathematical description of this metric to be satisfactory. It is our belief that this explanation would get away from the main topic of our article and we prefer to refer interested readers to the relevant publication by Zhang et al (2011).

For an in-depth description of the FSIM metric we would like to recommend the reading of the article in which it is first introduced[1]. This article is cited in our manuscript.

The authors of this article have made available the MATLAB source code of this metric[2].

As far as our work is concerned, we have used an existing python implementation of this same metric
* * *
[1] Zhang, L., Zhang, L., Mou, X., and Zhang, D.: FSIM: A Feature Similarity Index for Image Quality Assessment, IEEE Transactions on Image Processing, 20, 2378–2386, https://doi.org/10.1109/TIP.2011.2109730, 2011.

[2] https://web.comp.polyu.edu.hk/cslzhang/IQA/FSIM/FSIM.htm

provided by the piq library[3].

1. *Table 1: How many epochs were used in training? Please provide some samples of training/test losses over epoch to check the convergence/overfitting of the model.*

We have added an appendix on the convergence of our models, containing two graphics describing respectively the evolution of the loss function values during training, and the distribution of MAE values among test samples.

"**Figure 1** [Figure D1 in our manuscript] **provides a graph of training loss values over iterations, showing clearly the convergence of the model. This corresponds to the training of a UNet model using exlcusively the AOD as input. In this experiment as in all other experiments presented in this article, the models are trained on 500 epochs.**

**Figure 2** [Figure D2 in our manuscript] **gives, for the same model, an overview of the MAE values for the different test samples. A few test samples stand out as having a significantly worse MAE than others, but the maximum MAE for these samples remains below** $\mu g/m^3$ **, which is satisfying.**"

1. *Line 439: the inference time increases?*
2. *Line 576: are specifcs to –¿ are specific to*
3. *Line 590: However, and while –¿ However, while*
4. *Line 594: As suggested by (Zhou et al., 2024) –¿ As suggested by Zhou et al. (2024)*

Each of these typos and formulations have been corrected as suggested in the new version of our manuscript.
* * *
[3]https://piq.readthedocs.io/en/latest/

[Figure]

Figure 1: Graph of training loss during supervised learning over iterations

[Figure]

Figure 2: Graph of MAE values during testing over sample date

**Dear Second Reviewer,**

We would like to thank the reviewer for his insightful observations and comments. Some of the suggested modifications were applied to a new version of this manuscript while justification is provided hereafter otherwise.

1. *It is not clear how the relative versions of MAE and MBE (rMAE and rMBE) are defined. Please provide explicit definitions with mathematical formulations.*

In our new version mathematical formulations of the rMAE and rMBE are now provided.

1. *The summarizing scores (total score, timeless score, reduced score) undermine the purpose of having multiple error metrics. The different versions of summarizing scores also contradict the claim of easier model comparisons. Consequently, performance tables (e.g., table 2) are cluttered with both the 5 error metrics and additional 3 scores. I suggest prioritizing a subset of these metrics or scores to avoid confusion and overwhelming the readers.*

We would like to specify that we do agree with the referee that these scores may not be suited for an in-depth analysis of our results, and may make the analysis of our results harder. These scores are mainly used to justify the selection of some of our experiments among the numerous we have performed before comparing them, and provide a quantifiable rationale for this selection. Once this subset of experiments is selected, a more in-depth analysis can be realised based on dedicated metrics. Therefore we decided to only display the values of our metrics in the revised manuscript, and have updated the description of our scores accordingly. However our selection protocol, using our scores, remains the same. Our total score is also used in our radar charts to highlight the overall best results.

1. *The rationale behind the normalization of inference time in equation 5 needs clarification. Mixing inference time with other error metrics adds to the confusion. Consider presenting the inference time as a standalone performance metric and eliminating the total score.*

The rationale behind the choice of the 0.05s threshold for the normalization of inference time in equation 5 has been clarified in our new version.

"**Regarding the inference time, the threshold of** $0.05s$ **is used as it is the maximum inference time among our experiences with our deep learning models.**" (section 5.4)

Regarding the use of the total score, modifications have been made and are detailed in our response to the second comment.

1. *The definitions of the total score, timeless score, and reduced score require justification. While it is clear that QE, MAE, MBE, and FSIM are simple equally weighted average scores, their distribution (or typical range) and sensitivity are not the same. A simple average may assign higher weights to the most sensitive score.*

As stated in our answer to the second comment, these scores are not meant to be used as a way to analyse our results, but simply to provide a clearly defined protocol to select a subset of our results for analysis. Only the metrics should be used in the results analysis itself. This has now been made clearer in our new version of the manuscript.

1. *It is not straightforward to compare the same metric across different models using radar charts. This design highlights differences between the metrics of the same model.*

We do understand the referee's opinion regarding our choice to use radar charts to display our results overviews. However we failed to find a more satisfactory way to represent this amount of experimental results in a brief way. We do want to bring the attention of the referee to the fact that to realise this graph, all metrics have been normalized, the same way they were for the computation of the scores. This allows us to represent all metrics on the same value scale. This has now been made clearer in the new version of our manuscript.

1. *The values of the metrics are challenging to read from the plots. I suggest annotating each entry on the charts.*

Some of the entries on our radar charts are now annotated. We chose to annotate only the entries corresponding to the experiments which lead to the best results, as annotating all entries would have hindered the readability of these graphs.

1. *The common legend in figure 8 is counterintuitive, though it is understood that this was done to save space. The line styles and colors are simple enough to be described in captions.*

The common legend has been replaced with a paragraph describing the different colors and styles of the lines, in section 6.1, and a summary of this paragraph is present in the captions of our radar charts.

**To both referees,**

We hope to have adequately answered your concerns and thank you again for your interest in our work.

Kind regards,

**Matthieu Dabrowski**